# Efficient Codebook and Factorization for Second Order Representation Learning

## Abstract

Learning rich and compact representations is an open topic in many fields such as word embedding, visual question-answering, object recognition or image retrieval. Although deep neural networks (convolutional or not) have made a major breakthrough during the last few years by providing hierarchical, semantic and abstract representations for all of these tasks, these representations are not necessary as rich as needed nor as compact as expected. Models using higher order statistics, such as bilinear pooling, provide richer representations at the cost of higher dimensional features. Factorization schemes have been proposed but without being able to reach the original compactness of first order models, or at a heavy loss in performances. This paper addresses these two points by extending factorization schemes to codebook strategies, allowing compact representations with the same dimensionality as first order representations, but with second order performances. Moreover, we extend this framework with a joint codebook and factorization scheme, granting a reduction both in terms of parameters and computation cost. This formulation leads to state-of-the-art results and compact second-order models with few additional parameters and intermediate representations with a dimension similar to that of first-order statistics.

## 1 Introduction

Learning rich and compact representations is an open topic in many fields such as word embedding (Mikolov et al. (2013)), visual question-answering (Yang et al. (2016)), object recognition (Szegedy et al. (2015)) or image retrieval (Opitz et al. (2017)). The standard approach extracts features from the input data (text, image, *etc.*) and builds a representation that will be next processed for a given task (classification, retrieval, *etc.*). These features are usually extracted with deep neural networks and the representation is trained in an end-to-end manner. Recently, representations that compute first order statistics over input data have been outperformed by improved models that compute higher order statistics such as bilinear models. This embedding strategy generates richer representations and has been applied in a wide range of tasks : word embedding (Clinchant & Perronnin (2013)), VQA (Kim et al. (2017)), fine grained classification (Wei et al. (2018)), *etc.* and gets state-of-the-art results. For instance, Bilinear models perform the best for fine grained visual classification tasks by producing efficient representations that model more details within an image than classical first order statistics (Lin et al. (2015)).

However, even if the increase in performances is unquestionable, second order models suffer from a collection of drawbacks: Their intermediate dimension increases quadratically with respect to input features dimension, they require a projection to lower dimension that is costly both in number of parameters and in computation, they are harder to train than first order models due to the increased dimension, they lack a proper adapted pooling scheme which leads to sub-optimal representations.

The two main downsides, namely the high dimensional output representations and the sub-efficient pooling scheme, have been widely studied over the last decade. On one hand, the dimensionality issue has been studied through factorization scheme, either representation oriented such as Compact Bilinear Pooling (Gao et al. (2016)) and Hadamard Product for Low Rank Bilinear Pooling (Kim et al. (2017)), or task oriented as Low-rank Bilinear Pooling (Kong & Fowlkes (2017)). While these factorization schemes are efficient in term of computation cost and number of parameters, the

intermediate representation is still too large (typically 10k dimension) to ease the training process and using lower dimension greatly deteriorate performances.

On the other hand, it is well-known that global average pooling schemes aggregate unrelated features. This problem has been tackled by the use of codebooks such as VLAD (Arandjelovic & Zisserman (2013)) or, in the case of second-order information, Fisher Vectors (Perronnin et al. (2010)). These strategies have been enhanced to be trainable in an end-to-end manner (Arandjelovic et al. (2016); Tang et al. (2016)). However, using a codebook on end-to-end trainable second order features leads to an unreasonably large model, since the already large second order model has to be duplicated for each entry of the codebook. This is for example the case in MFAFVNet (Li et al. (2017b)) for which the second order layer alone (*i.e.*, without the CNN part) already costs over 25M parameters and 40 GFLOP, or about as much as an entire ResNet50.

In this paper, we tackle both of these shortcomings (intermediate representation cost and lack of proper pooling) by exploring joint factorization and codebook strategies. Our main results are the following:

- We first show that state-of-the-art factorization schemes can already be improved by the use of a codebook pooling, albeit at a prohibitive cost.
- We then propose our main contribution, a joint codebook and factorization scheme that achieves similar results at a much reduced cost.

Since our approach focuses on representation learning and is task agnostic, we validate it in a retrieval context on several image datasets to show the relevance of the learned representations. We show our model achieves competitive results on these datasets at a very reasonable cost.

The remaining of this paper is organized as follows: in the next section, we present the related work on second order pooling, factorization schemes and codebook strategies. In section 3, we present our factorization with the codebook strategy and how we improve its integration. In section 4, we show an ablation study on the Stanford Online Products dataset (Oh Song et al. (2016)). Finally, we compare our approach to the state-of-the-art methods on three image retrieval datasets (Stanford Online Products, CUB-200-2001, Cars-196).

## 2 RELATED WORK

In this section, we focus on methods that use representations based on second-order information and we provide a comparison in terms of computational efficiency and number of parameters for the second-order layer. These second-order methods exploit either bilinear pooling (section 2.1) and factorization schemes (section 2.2) or codebook strategies (section 2.3).

### 2.1 SECOND-ORDER POOLING

In this section, we briefly review end-to-end trainable Bilinear pooling (Lin et al. (2015)). This method extracts representations from the same image with two CNNs and computes the cross-covariance as representation. This representation outperforms its first-order version and other second-order representations such as Fisher Vectors (Perronnin et al. (2010)) once the global architecture is fine-tuned. However, bilinear pooling leads to a small improvement compared to second-order pooling (*i.e.*, the covariance of the CNN features) at the cost of a higher computation. Most of recent works on bilinear pooling only focus on computing covariance of the extracted features, that is :

$$\boldsymbol{y} = \sum_{i \in \mathbb{S}} \boldsymbol{x}_i \boldsymbol{x}_i^T = \boldsymbol{X}\boldsymbol{X}^T \in \mathbb{R}^{d \times d} \tag{1}$$

where $\boldsymbol{X} = \{\boldsymbol{x}_i \in \mathbb{R}^d | i \in \mathbb{S}\} \in \mathbb{R}^{d \times hw}$ is the matrix of concatenated CNN features, $h$ and $w$ are the height and the width of the extracted feature map. Another formulation is the vectorized version of $\boldsymbol{y}$ obtained by computing the Kronecker product ($\otimes$) of $\boldsymbol{x}_i$ with itself:

$$\boldsymbol{y} = \sum_{i \in \mathbb{S}} \boldsymbol{x}_i \otimes \boldsymbol{x}_i = \text{vec}(\boldsymbol{X}\boldsymbol{X}^T) \in \mathbb{R}^{d^2} \tag{2}$$

Due to the very high dimension of this representation, most recent works on bilinear pooling tend to improve this representation by providing new factorization scheme to reduce the computation.

## 2.2 Factorization schemes

Recent works on bilinear pooling proposed factorization schemes with two objectives: avoiding the direct computation of second order features and reducing the high dimensionality output representation. Gao et al. (2016) proposed Compact Bilinear Pooling that tackles the high dimensionality of second-order features by analyzing two low-rank approximations of the equivalent polynomial kernel (equation (2) from Gao et al. (2016)):

$$\langle \mathcal{B}(\mathcal{X}) \; ; \; \mathcal{B}(\mathcal{Y}) \rangle = \sum_{x_s \in \mathcal{B}(\mathcal{X})} \sum_{y_u \in \mathcal{B}(\mathcal{Y})} \langle x_s \; ; \; y_u \rangle^2 \approx \left\langle \sum_{x_s \in \mathcal{B}(\mathcal{X})} \phi(x_s) \; ; \; \sum_{y_u \in \mathcal{B}(\mathcal{Y})} \phi(y_u) \right\rangle \quad (3)$$

with $\phi$ the mapping function that approximates the kernel. This compact formulation allows to keep less than 4% of the components with nearly no loss in performances compared to the uncompressed model. Kim et al. (2017) improved Compact Bilinear Pooling using Hadamard Product and generalized it for visual question-answering tasks.

Kong & Fowlkes (2017) introduced Low Rank Bilinear Pooling (LR-BP) that takes advantage of SVM formulation by jointly training the network and the classifier. The authors propose an efficient factorization as they never compute directly the covariance features and have slightly better results compared to the uncompressed bilinear pooling or Compact Bilinear Pooling. However, as it is, their method is limited to classification with the SVM formulation and cannot be used for other tasks.

Li et al. (2017a) introduced Factorized Bilinear Network (FBN), an improved version of Compact Bilinear Pooling. FBN never directly computes the covariance matrix. Instead, it projects the features using a hyperplan and computes a quadratic form. The matrix of this quadratic form is supposed rank deficient to reduce the number of parameters and the computation cost with negligible loss in performances. Thus, the output representation (or directly the number of classes for classification tasks) is generated by concatenating these scalars for all projections.

Wei et al. (2018) presented Grassmann Bilinear Pooling as a new factorization scheme. The objective is to take advantage of the rank deficient covariance matrix using Singular Value Decomposition (SVD) and then compute the classifier over these Grassmann manifolds. This factorization is efficient in the sense that it never directly computes the second-order representation and contrary to LR-BP, this formulation allows the construction of a representation, by replacing the number of classes by the representation dimension. In practice, however, they need to greatly reduce the input feature dimension due to the SVD complexity which is cubic in the feature dimension.

In this work, we start from a similar factorization as Kim et al. (2017) detailed in section 3.1. However, this factorization is improved by the introduction of a codebook strategy that allows smaller representation dimension and improves performances.

## 2.3 Codebook strategies

An acknowledged drawback of pooling methods is that they pool unrelated features that may decrease performances. To cope with this observation, codebook strategies have been proposed and greatly improved performances by pooling only features that belong to the same codeword.

The first representations that take advantage of codebook strategies are Bag of Words (BoW) and in the case of second order information Fisher Vectors (Perronnin et al. (2010)). Fisher Vectors (FVs) extend the BoW framework by replacing the hard assignment of BoW by a Gaussian Mixture Model (GMM) and then compute the representation as an extension of the Fisher Kernel. In practice, covariance matrices are supposed to be diagonal which leads to representations of size $N(2d + 1)$ where $d$ is the dimension of the features and $N$ is the codebook size. Tang et al. (2016) proposed FisherNet, an architecture that integrates FVs as differentiable layer. The proposed layer outperforms non-trainable FVs approach but nonetheless has the high output dimension of the original FV.

Li et al. (2017b) introduced MFA-FV network, a deep architecture which extends the MFA-FV of Dixit & Vasconcelos (2016) by producing a second order information embedding trainable in an end-to-end manner. The proposed formulation takes advantage of both worlds: MFA-FV generates an efficient representation of non-linear manifolds with a small latent space and it can be trained in an end-to-end manner. The main drawbacks of their method is the direct computation of second-order

| Method | C. | F. | #param | computation | dim. |
|---|---|---|---|---|---|
| BP | $\times$ | $\times$ | - | $hwd^2$ [206M] | $d^2$ [262k] |
| CBP-RM[†] | $\times$ | $\checkmark$ | $2dD$ [10M] | $2hwdD$ [8G] | $D$ [10k] |
| CBP-TS[†] | $\times$ | $\checkmark$ | $2d$ [1k] | $hw(d + 3D \log D)$ [94M] | $D$ [10k] |
| HPBP | $\times$ | $\checkmark$ | $2dD$ | $2hwdD$ | $D$ |
| FBN | $\times$ | $\checkmark$ | $dkD$ [5M] | $hwdkD$ [4.1G] | $D$ [512] |
| Grassmann BP | $\times$ | $\checkmark$ | $LdD$ [4.2M] | $c^3 + DLc^2$ [2.3G] | $D$ [512] |
| MFAFVNet[†] | $\checkmark$ | $\times$ | $N(d^2 + dL)$ [27M] | $Nd^2(2P + L)$ [42G] | $NDL$ [500k] |
| Ours | $\checkmark$ | $\checkmark$ | $2RdD$ [4.2M] | $2hwdDR$ [3.2G] | $D$ [512] |

Table 1: Summary of second order methods. C. and F. columns are respectively for "Codebook" and "Factorization". Numbers in brackets are typical values. Methods marked [†] used the original paper values. Other methods use the following parameters: $h = w = 28$ are the height and width of the feature map, $d = 512$ is the feature dimension, $D$ is the output representation, and is set to 512 if possible. Our proposed method uses a codebook $N = 32$ and a projection set of size $R = 8$.

features for each codeword (computation cost), the raw projection of this covariance matrix into the latent space for each codeword (computation cost and number of parameters), and finally the representation dimension. In the original paper, the proposed representation reaches 500k dimension, twice the already high dimension of Bilinear Pooling.

For a more compact review, computation cost, number of parameters, use of codebook and/or factorization are sumed-up in table 1. This table shows that, to our knowledge, no efficient factorization combined with codebook strategy has been proposed to exploit the richer representation due to codebook but at a small increase in terms of number of parameters or computation cost. As is shown in this table, our proposition combine the best of both worlds by providing a joint codebook and factorization optimization scheme with a similar number of parameters and computation cost to that of methods without codebook strategies.

## 3 METHOD OVERVIEW

In section 3.1, we detail the initial factorization scheme and the properties of the Kronecker Product and the dot product that are used in the two next sections. In section 3.2, we extend this factorization to a codebook strategy and show the limitations of this architecture in terms of computation cost, low-rank approximation, number of parameters, *etc*. In section 3.3 we enhance this representation by sharing projectors to all codewords into the codebook, leading to a joint codebook and factorization optimization.

### 3.1 INITIAL FACTORIZATION SCHEME

In this section, we present the factorization used and highlight the advantages and limitations of this scheme. For a given input feature $\boldsymbol{x} \in \mathbb{R}^d$, we compute its second-order representation $\boldsymbol{x} \otimes \boldsymbol{x} \in \mathbb{R}^{d^2}$ and project it into a smaller subspace with $\boldsymbol{W} \in \mathbb{R}^{d^2 \times D}$ to build the output feature $\boldsymbol{z}(\boldsymbol{x}) \in \mathbb{R}^D$. These output features are then pooled to build the output representation $\boldsymbol{z}$:

$$\boldsymbol{z} = \sum_{\boldsymbol{x}} \boldsymbol{z}(\boldsymbol{x}) = \sum_{\boldsymbol{x}} \boldsymbol{W}^T(\boldsymbol{x} \otimes \boldsymbol{x}) \tag{4}$$

In the rest of the paper, we use the notation $z_i$ that refers to the $i$-th dimension of the output representation $\boldsymbol{z}$ and $z_i(\boldsymbol{x})$ the $i$-th dimension of the output feature $\boldsymbol{z}(\boldsymbol{x})$, that is:

$$z_i = \sum_{\boldsymbol{x}} z_i(\boldsymbol{x}) = \sum_{\boldsymbol{x}} \boldsymbol{w}_i^T(\boldsymbol{x} \otimes \boldsymbol{x}) = \sum_{\boldsymbol{x}} \langle \boldsymbol{w}_i \; ; \; \boldsymbol{x} \otimes \boldsymbol{x} \rangle \tag{5}$$

with $\boldsymbol{w}_i \in \mathbb{R}^{d^2}$ a column of $\boldsymbol{W}$ and $\langle \cdot \; ; \; \cdot \rangle$ the dot product. Then we enforce a factorization of $\boldsymbol{w}_i$ to take advantage of the properties of dot product and Kronecker product, that is $\forall (\boldsymbol{a}, \boldsymbol{b}, \boldsymbol{c}, \boldsymbol{d}) \in (\mathbb{R}^d)^4$ , $\langle \boldsymbol{a} \otimes \boldsymbol{c} \; ; \; \boldsymbol{b} \otimes \boldsymbol{d} \rangle = \langle \boldsymbol{a} \; ; \; \boldsymbol{b} \rangle \langle \boldsymbol{c} \; ; \; \boldsymbol{d} \rangle$. Thus, we use the following rank one decomposition of $\boldsymbol{w}_i = \boldsymbol{u}_i \otimes \boldsymbol{v}_i$ where $(\boldsymbol{u}, \boldsymbol{v}) \in (\mathbb{R}^d)^2$. $z_i(\boldsymbol{x})$ from equation (5) becomes:

$$z_i(\boldsymbol{x}) = \langle \boldsymbol{u}_i \; ; \; \boldsymbol{x} \rangle \langle \boldsymbol{v}_i \; ; \; \boldsymbol{x} \rangle \tag{6}$$

This factorization is efficient in term of parameters as it needs only $2dD$ parameters instead of $d^2D$ for the full projection matrix. However, even if this rank one decomposition allows interesting dimension reduction (Gao et al. (2016); Kim et al. (2017)) it is not enough to keep rich representation with smaller dimension. Consequently, we extend the second-order feature to a codebook strategy.

## 3.2 CODEBOOK STRATEGY

To extend second-order pooling, we want to pool only similar features, that is which belong to the same codeword. This codebook pooling is interesting because each projection to a sub-space should have only similar features, and they should be encoded with fewer dimension. For a codebook size of $N$, we compute an assignment function $\boldsymbol{h}(\cdot) \in \mathbb{R}^N$. This function could be a hard assignment function (*e.g.*, $\arg \min$ over distance to each cluster) or a soft assignment (*e.g.*, the softmax function). Thus, our output feature $z_i(\boldsymbol{x})$ becomes:

$$z_i(\boldsymbol{x}) = \langle \boldsymbol{w}_i \; ; \; \boldsymbol{h}(\boldsymbol{x}) \otimes \boldsymbol{x} \otimes \boldsymbol{h}(\boldsymbol{x}) \otimes \boldsymbol{x} \rangle \tag{7}$$

Remark that now $\boldsymbol{W} \in \mathbb{R}^{N^2 d^2 \times D}$ and $\boldsymbol{w}_i \in \mathbb{R}^{N^2 d^2}$. Here, we duplicate $\boldsymbol{h}(\boldsymbol{x})$ for generalization purpose: In the case of the original bilinear pooling this formulation becomes $\boldsymbol{h}_1(\boldsymbol{x}_1) \otimes \boldsymbol{x}_1 \otimes \boldsymbol{h}_2(\boldsymbol{x}_2) \otimes \boldsymbol{x}_2$ and we can use two different codebooks, one for each network. Moreover, this formulation allows more degrees of freedom for the next factorization. As in equation 6, we enforce the rank one decomposition of $\boldsymbol{w}_i = \boldsymbol{p}_i \otimes \boldsymbol{q}_i$ where $(\boldsymbol{p}_i, \boldsymbol{q}_i) \in (\mathbb{R}^{Nd})^2$. This first factorization leads to the following output feature $z_i(\boldsymbol{x})$:

$$z_i(\boldsymbol{x}) = \langle \boldsymbol{p}_i \; ; \; \boldsymbol{h}(\boldsymbol{x}) \otimes \boldsymbol{x} \rangle \langle \boldsymbol{q}_i \; ; \; \boldsymbol{h}(\boldsymbol{x}) \otimes \boldsymbol{x} \rangle \tag{8}$$

This intermediate representation is too large to be computed directly, *e.g.* using $N = 100$ and the same parameters as in Table 1, we have intermediate features with 50k dimensions and the two intermediate feature maps consume about 320MB of memory which becomes rapidly intractable if the dimension of $\boldsymbol{z}$ is above 10. Then, we enforce two other factorizations of $\boldsymbol{p}_i = \sum_j \boldsymbol{e}^{(j)} \otimes \boldsymbol{u}_{i,j}$ and $\boldsymbol{q}_i = \sum_j \boldsymbol{e}^{(j)} \otimes \boldsymbol{v}_{i,j}$ where $\boldsymbol{e}^{(j)} \in \mathbb{R}^N$ is the $j$-th vector from the natural basis of $\mathbb{R}^N$ and $(\boldsymbol{u}_{i,j}, \boldsymbol{v}_{i,j}) \in (\mathbb{R}^d)^2$. Then equation (8) becomes:

$$z_i(\boldsymbol{x}) = \left( \sum_{j=1}^N \left\langle \boldsymbol{h}(\boldsymbol{x}) \; ; \; \boldsymbol{e}^{(j)} \right\rangle \langle \boldsymbol{x} \; ; \; \boldsymbol{u}_{i,j} \rangle \right) \left( \sum_{j=1}^N \left\langle \boldsymbol{h}(\boldsymbol{x}) \; ; \; \boldsymbol{e}^{(j)} \right\rangle \langle \boldsymbol{x} \; ; \; \boldsymbol{v}_{i,j} \rangle \right) \tag{9}$$

The decompositions of $\boldsymbol{p}_i$ and $\boldsymbol{q}_i$ play similar roles as intra-projection in VLAD representation (Delhumeau et al. (2013)). Indeed, if we consider $\boldsymbol{h}(\cdot)$ as a hard assignment function, the projection that will be computed is the only one assigned to the corresponding codewords. Thus, this model learns a projection matrix for each codebook entry.

Furthermore, equation (9) can be factorized using $h_j(\boldsymbol{x})$, the $j$-th component of $\boldsymbol{h}(\boldsymbol{x})$:

$$
\begin{aligned}
z_i(\boldsymbol{x}) &= \left( \sum_{j=1}^N h_j(\boldsymbol{x}) \langle \boldsymbol{x} \; ; \; \boldsymbol{u}_{i,j} \rangle \right) \left( \sum_{j=1}^N h_j(\boldsymbol{x}) \langle \boldsymbol{x} \; ; \; \boldsymbol{v}_{i,j} \rangle \right) \\
&= \left( \boldsymbol{h}(\boldsymbol{x})^T \boldsymbol{U}_i^T \boldsymbol{x} \right) \left( \boldsymbol{h}(\boldsymbol{x})^T \boldsymbol{V}_i^T \boldsymbol{x} \right)
\end{aligned}
\tag{10}
$$

where $\boldsymbol{U}_i \in \mathbb{R}^{d \times N}$ is the matrix concatenating the projections of all entries of the codebook for the $i$-th output dimension. We call this approach *C-CBP* as it corresponds to the extension of CBP (Gao et al. (2016)) to a *Codebook* strategy.

This representation has multiple advantages: First, it computes second order features that leads to better performances compared to its first order counterpart. Second, the first factorization provides an efficient alternative in terms of number of parameters and computation despite the decreasing performances when it reaches small representation dimension. This downside is addressed by the third advantage which is the codebook strategy. It allows the pooling of only related features while their projections to a sub-space is more compressible. However, even if this codebook strategy improves the performances, the number of parameters is in $\mathcal{O}(dDN)$ As such, using large codebook may become intractable. In the next section, we extend this scheme by sharing a set of projectors and enhance the decompositions of $\boldsymbol{p}_i$ and $\boldsymbol{q}_i$.

## 3.3 SHARING PROJECTORS

In the previous model, for a given codebook entry, there is one dedicated projector that is learned to map to a smaller vector space all features that belong to this codebook entry. The proposed idea is, instead of using a a one-to-one correspondence, we learn a set of projectors that is shared across the codebook. The reasoning behind is that projectors from different codebook entries are unlikely to be all orthogonal. By doing such hypothesis, that is, the vector space spaned by the projection matrices has a lower dimension than the codebook itself, we can have smaller models with nearly no loss in performances. To check this hypothesis, we extend the proposed factorization from section 3.2. We want to generate $\boldsymbol{U}_i$ from $\{\widetilde{\boldsymbol{U}}_i\}_{i\in\{1,...,R\}}$ and $\boldsymbol{V}_i$ from $\{\widetilde{\boldsymbol{V}}_i\}_{i\in\{1,...,R\}}$ where $R$ is the number of projections in the set. Then the two new enforced factorization of $\boldsymbol{p}_i$ and $\boldsymbol{q}_i$ are:

$$\boldsymbol{p}_i(\boldsymbol{x}) = \sum_r f_{p,r}\Big(\boldsymbol{h}(\boldsymbol{x})\Big)\boldsymbol{e}^{(r)} \otimes \widetilde{\boldsymbol{u}}_{i,r} \;\; \text{and} \;\; \boldsymbol{q}_i(\boldsymbol{x}) = \sum_r f_{q,r}\Big(\boldsymbol{h}(\boldsymbol{x})\Big)\boldsymbol{e}^{(r)} \otimes \widetilde{\boldsymbol{v}}_{i,r} \qquad (11)$$

where $\boldsymbol{f}_p$ and $\boldsymbol{f}_q$ are two functions from $\mathbb{R}^N$ to $\mathbb{R}^R$ that transform the codebook assignment into a set of coefficient which generate their respective projection matrices. Then, using these factorizations lead to the following equation:

$$\begin{aligned} z_i(\boldsymbol{x}) &= \left(\sum_{r=1}^{R} f_{p,r}\Big(\boldsymbol{h}(\boldsymbol{x})\Big)\langle\boldsymbol{x}\;;\;\widetilde{\boldsymbol{u}}_{i,j}\rangle\right)\left(\sum_{r=1}^{R} f_{q,r}\Big(\boldsymbol{h}(\boldsymbol{x})\Big)\langle\boldsymbol{x}\;;\;\widetilde{\boldsymbol{v}}_{i,j}\rangle\right) \\ &= \left(\boldsymbol{f}_p\Big(\boldsymbol{h}(\boldsymbol{x})\Big)^T\widetilde{\boldsymbol{U}}_i^T\boldsymbol{x}\right)\left(\boldsymbol{f}_q\Big(\boldsymbol{h}(\boldsymbol{x})\Big)^T\widetilde{\boldsymbol{V}}_i^T\boldsymbol{x}\right) \end{aligned} \qquad (12)$$

In this paper, we only study the case of a linear projection to the sub-space $\mathbb{R}^R$, that is $\boldsymbol{f}_p : \boldsymbol{a} \mapsto \boldsymbol{f}_p(\boldsymbol{a}) = \boldsymbol{A}^T\boldsymbol{a}$ and $\boldsymbol{f}_q : \boldsymbol{a} \mapsto \boldsymbol{f}_q(\boldsymbol{a}) = \boldsymbol{B}^T\boldsymbol{a}$ with $(\boldsymbol{A}, \boldsymbol{B}) \in (\mathbb{R}^{N\times R})^2$. Finally, the fully factorized $\boldsymbol{z}$ transform is computed using the following equation:

$$z_i(\boldsymbol{x}) = \left(\boldsymbol{h}(\boldsymbol{x})^T\boldsymbol{A}\widetilde{\boldsymbol{U}}_i^T\boldsymbol{x}\right)\left(\boldsymbol{h}(\boldsymbol{x})^T\boldsymbol{B}\widetilde{\boldsymbol{V}}_i^T\boldsymbol{x}\right) \qquad (13)$$

Equation (13) is more efficient in terms of parameters than equation (10) as it requires only $2(RdD + NR)$ parameters instead of $2NdD$. We call this approach *JCF* for *Joint Codebook and Factorization*. This shared projection is both efficient in terms of number of parameters and in computation by a factor $R/N$. In the next section, we provide an ablation study of the proposed method, comparing equation (10) and equation (13), demonstrating that learning recombination is both efficient and performing.

## 3.4 IMPLEMENTATION DETAILS

In this section, we give some details about our implementation. All our experiments are performed on image retrieval datasets to assess the quality of the representations independantly of any classification scheme. We build our model over pre-trained network such as VGG16 (Simonyan & Zisserman (2014)) or ResNet50 (He et al. (2016)). In both case we reduce the features dimension to 256 dimensions and we $l_2$-normalize them. For the assignment function $\boldsymbol{h}$, we use the softmax over cosine similarity between the features and the codebook. Once the second-order representations are computed we pull them using global average pooling and we $l_2$-normalize the output representation. Similarities between images are computed using the cosine similarity. In metric, we use Recall@K which takes the value 1 if there is at least one element from the same instance in the top-K results else 0 and averages these scores over the test set. The network is trained in 3 steps using a standard triplet loss function. In the first step, we freeze the ResNet50 and only train our added layers with 100 images per batch, we sample the negative within the batch and we use a learning rate of $10^{-4}$ for 40 epochs. In the second one, we unfreeze ResNet50 and fine-tune the whole architecture for 40 epochs more with a learning rate of $10^{-5}$ and a batch of 64 images with the negative sampled within the batch. In the last one, we fine-tune the network with a batch size of 64 images sampled by hard mining the training set with a learning rate of $10^{-5}$. The margin of the triplet loss is set to 0.1. Images are resized to 224x224 pixels for both train and test sets.

| Method | Baseline | BP | C-BP | CBP | C-CBP | C-CBP | C-CBP |
|--------|----------|-----|------|-----|-------|-------|-------|
| Codebook | - | - | 4 | - | 4 | 16 | 32 |
| Parameters | 1M | 34M | 135M | 0.8M | 1.6M | 4.7M | 8.9M |
| R@1 | 63.8 | 65.9 | 67.1 | 64.7 | 66.4 | 68.1 | 70.2 |

Table 2: Comparison of codebook strategy in terms of parameters and performances between a classical retrieval model, standard bilinear pooling and factorization using Hadamard product.

| Codebook size | 32 | | | | 16 | | |
|---------------|-----|-----|-----|-----|-----|-----|-----|
| Rank | 32 | 16 | 8 | 4 | 16 | 8 | 4 |
| R@1 | 70.6 | 69.7 | 69.4 | 68.1 | 69.8 | 68.3 | 68.2 |

Table 3: Recall@1 for different combination of JCF factorization.

# 4 ABLATION STUDIES

## 4.1 BILINEAR POOLING AND CODEBOOK STRATEGY

In this section, we demonstrate both the relevance of second-order information for retrieval tasks and the influence of the codebook on our method. We report recall@1 on Stanford Online Products in Table 2 for the different configuration detailed below with the training procedure from Section 3.4 without the hard mining step.

First, as a reference, we train a ResNet50 with a global average pooling and a fully connected layer to project the representation dimension from 2048 to 512. We denote it *Baseline*. Then we re-implement Bilinear Pooling (BP) and Compact Bilinear Pooling (CBP) and extend them naively to a codebook strategy (C-BP and C-CBP). The objective is to demonstrate that such strategy performs well, but a an intractable cost. Results are reported in Table 2. Note that, for each bilinear pooling method, we first add a $1 \times 1$ convolution to project the ResNet50 features from 2048 to 256 dimensions. This experiment confirm the interest of bilinear pooling in image retrieval with a improvement of 2% over the baseline, while using a 512 dimension representation. Furthermore, even using a codebook strategy with few codewords enhance bilinear pooling by 1% more, however, the number of parameters become intractable for codebook of size greater than 4: this naive strategy requires 270M parameters to extend this model to a codebook with a size of 8.

Using the factorization from equation (10) greatly reduces the required number of parameters and allows the exploration of larger codebook. However, this factorization without codebook leads to lower scores than the non factorized bilinear pooling, but adding a codebook strategy increases performances by more than 4% over bilinear pooling without codebook, with nearly 4 times less parameters.

## 4.2 SHARING PROJECTIONS

In this part, we study the impact of the sharing projection. We use the same training procedure as in the previous section. For each codebook size, we train architecture with a different number of projections, allowing to compare architectures without the sharing process to architectures with greater codebook size but with the same number of parameters by sharing projectors. Results are reported in Table 3. Sharing projectors leads to smaller models with few loss in performances, and using richer codebooks allows more compression with superior results.

# 5 COMPARISON WITH BILINEAR FACTORIZATION

In this section, we report performances of our factorization on 3 fine-grained visual classification (FGVC) datasets: CUB (Wah et al. (2011)), CARS (Krause et al. (2013)) and AIRCRAFT (Maji et al. (2013)). We use VGG16 as backbone network. Furthermore, to demonstrate the effectiveness of our codebook based factorization scheme to produce compact but effective second-order representations we compare JCF to closely-related formulations on FGVC tasks, that are:
**HPBP** ($z_i(\boldsymbol{x}) = \boldsymbol{p}_i^T[\sigma(\boldsymbol{U}^T\boldsymbol{x}) \odot \sigma(\boldsymbol{V}^T\boldsymbol{x})]$). Non-linear multi-rank with shared $\boldsymbol{U}, \boldsymbol{V}$.

| Method | CUB | CARS | AIRCRAFT | Feature dim. | Parameters |
|---|---|---|---|---|---|
| Full BP - Lin et al. (2015) | 84.1 | 90.6 | 86.9 | 256k | 200MB |
| CBP-RM - Gao et al. (2016) | 83.9 | 90.5 | 84.3 | 8192 | 38MB |
| CBP-TS - Gao et al. (2016) | 84.0 | 91.2 | 84.1 | 8192 | 6.3MB |
| MoNet - Gou et al. (2018) | 85.7 | 90.8 | 88.1 | 10k | 4KB |
| LRBP - Kong & Fowlkes (2017) | 84.2 | 90.9 | 87.3 | 10k | 0.8MB |
| FBP - Li et al. (2017a) | 82.9 | - | - | - | - |
| SMSO - Yu & Salzmann (2018) | 85.0 | - | - | 2048 | 0.06MB |
| HPBP* | 82.6 | 89.4 | 86.6 | 2048 | |
| MR | 83.1 | 89.0 | 85.9 | 512 | 8MB |
| MR+NL | 82.3 | 89.4 | 85.5 | 512 | |
| MR+NL+C | 82.4 | 89.5 | 86.5 | 512 | |
| JCF ($N = 32, R = 8$) | 84.3 | 90.4 | 87.3 | 512 | 8MB |

Table 4: Evaluation of our proposed factorization scheme. We compare our method to the state-of-the-art on Bilinear factorization and similar methods. We evaluate them with small representation dimension to attest our dimensionality reduction efficiency. * denotes our re-implementation.

**MR** ($z_i(\boldsymbol{x}) = \boldsymbol{x}^T \boldsymbol{U}_i \boldsymbol{V}_i^T \boldsymbol{x}$). Multi-rank extension of Eq.(6) which allows to compare the benefit of codebook against direct rank increase. This approach is related to FBP which uses a higher rank decomposition ($R = 20$) than in our tests ($R = 8$).
**MR+NL** ($z_i(\boldsymbol{x}) = \boldsymbol{1}_R^T [\sigma(\boldsymbol{U}_i^T \boldsymbol{x}) \odot \sigma(\boldsymbol{V}_i^T \boldsymbol{x})]$) Multi-rank with the same non-linearity as HPBP.
**MR+NL+C** ($z_i(\boldsymbol{x}) = \boldsymbol{p}^T [\sigma(\boldsymbol{U}_i^T \boldsymbol{x}) \odot \sigma(\boldsymbol{V}_i^T \boldsymbol{x})]$) which adds weights to the multi-rank combination. For a fair comparison, we fix the number of parameters for all of these methods to the same number as JCF ($N = 32, R = 8$). Thus, all methods use $d = 256$ and $D = 512$ and $R = 8$ except HPBP which uses $R = 2048$ to compensate for its shared matrices $\boldsymbol{U}, \boldsymbol{V}$.

We report classification accuracy on the three aforementioned datasets in Table 4. As we can see, our method consistently outperforms the multi-rank variants. This confirms our intuition about the importance of grouping features by similarity before projection and aggregation. Indeed, multi-rank variants do not have a selection mechanism preceding the projection into the subspace that would allow to selectively choose the projectors based on the input features. Instead, all features are projected using the same projectors and then aggregated. We argue that non-linear multi-rank variants bring only marginal improvements, since the non-linearity happens after the projection is made. Although it is still possible to learn a projection coupled with the non-linearity that would lead to a similarity driven aggregation, it is not enforced by design. Since JCF does the similarity driven aggregation by design, it is easier to train, which we believe explains the results.

## 6 Comparison to the state-of-the-art

In this section, we compare our method to the state-of-the-art on 3 retrieval datasets: Stanford Online Products (Oh Song et al. (2016)), CUB-200-2011 (Wah et al. (2011)) and Cars-196 (Krause et al. (2013)). For Stanford Online Products and CUB-200-2011, we use the same train/test split as Oh Song et al. (2016). For Cars-196, we use the same as Opitz et al. (2017). We report the standard recall@K with $K \in \{1, 10, 100, 1000\}$ for Stanford Online Products and with $K \in \{1, 2, 4, 8, 16, 32\}$ for the other two.

On CUB-200-2011 and Cars-196 (see Table 6) we re-implement Bilinear Pooling (BP) and Compact Bilinear Pooling (CBP) on a VGG16. Even if the results are interesting, the constraint over intermediate representation is too strong to achieve relevant results. We then implement the codebook factorization from equation 10 a codebook size of 32 (denoted C-CBP). This formulation outperforms both classical second-order models by a large margin with few parameters. Moreover, our model that shares projections over the codebook (JCF , computed following equation 13) with $R = 8$ has 4 times less parameters for a 2% loss on CUB-200-2011 dataset. On Cars-196, the sharing induces a higher loss, but may be improved with more projections to share.

On Stanford Online Products, we report the *Baseline*, implementations from equations 10 (C-CBP) and 13 (JCF ) with $R = 8$. We achieve state-of-the-art results using both methods and more than 10% improvement over the *Baseline*. Remark that JCF costs 4 times less than C-CBP at a 1% loss.

Table 5: Comparison with the state-of-the-art on Stanford Online Products dataset. * Denote our re-implementation. **bold** scores are the current state-of-the-art and underlined are the second ones.

| r@ | 1 | 10 | 100 | 1000 |
|---|---|---|---|---|
| LiftedStruct (Oh Song et al. (2016)) | 62.1 | 79.8 | 91.3 | 97.4 |
| Binomial Deviance (Ustinova & Lempitsky (2016)) | 65.5 | 82.3 | 92.3 | 97.6 |
| N-Pair-Loss (Sohn (2016)) | 67.7 | 83.8 | 93.0 | 97.8 |
| HDC (Yuan et al. (2016)) | 69.5 | 84.4 | 92.8 | 97.7 |
| Margin (Wu et al. (2017)) | 72.7 | 86.2 | 93.8 | 98.0 |
| BIER (Opitz et al. (2017)) | 72.7 | 86.5 | 94.0 | 98.0 |
| Proxy NCA (Movshovitz-Attias et al. (2017)) | 73.7 | - | - | - |
| HTL (Ge (2018)) | 74.8 | 88.3 | 94.8 | 98.4 |
| ResNet50 *Baseline* | 63.8 | 80.0 | 91.0 | 97.1 |
| ResNet50 + C-CBP ($D = 32$) | **77.4** | 89.9 | **95.8** | 98.6 |
| ResNet50 + JCF ($D = 32, R = 8$) | 76.6 | **90.0** | **95.8** | **98.7** |

Table 6: Comparison with the state-of-the-art on CUB-200-2011 and Cars-196 datasets. * Denote our re-implementation. **bold** scores are the current state-of-the-art and underlined are second.

| | CUB-200-2011 | | | | | | Cars-196 | | | | | |
|---|---|---|---|---|---|---|---|---|---|---|---|---|
| r@ | 1 | 2 | 4 | 8 | 16 | 32 | 1 | 2 | 4 | 8 | 16 | 32 |
| Binomial Deviance | 52.8 | 64.4 | 74.7 | 83.9 | 90.4 | 94.3 | - | - | - | - | - | - |
| N-Pair-Loss | 51.0 | 63.3 | 74.3 | 83.2 | - | - | 71.1 | 79.7 | 86.5 | 91.6 | - | - |
| HDC | 53.6 | 65.7 | 77.0 | 85.6 | 91.5 | 95.5 | 73.7 | 83.2 | 89.5 | 93.8 | 96.7 | 98.4 |
| Margin | **63.6** | **74.4** | **83.1** | **90.0** | **94.2** | - | 79.6 | 86.5 | 91.9 | 95.1 | 97.3 | - |
| BIER | 55.3 | 67.2 | 76.9 | 85.1 | 91.7 | 95.5 | 78.0 | 85.8 | 91.1 | 95.1 | 97.3 | 98.7 |
| HTL | 57.1 | 68.8 | 78.7 | 86.5 | 92.5 | 95.5 | 81.4 | 88.0 | 92.7 | 95.7 | 97.4 | **99.0** |
| VGG16 + BP* | 55.7 | 67.6 | 77.4 | 85.3 | 91.3 | 95.1 | 77.6 | 85.6 | 91.2 | 94.5 | 96.9 | 98.3 |
| VGG16 + CBP* | 53.7 | 66.1 | 76.0 | 84.8 | 91.2 | 95.4 | 75.1 | 83.8 | 89.3 | 93.9 | 96.5 | 98.3 |
| VGG16 + C-CBP ($D = 32$) | 60.1 | 72.1 | 81.7 | 88.3 | 93.4 | 96.4 | **82.6** | **89.2** | **93.5** | **96.0** | **97.8** | 98.9 |
| R50 + C-CBP ($D = 32$) | 60.1 | 71.9 | 82.4 | 89.4 | 93.9 | 96.8 | 79.5 | 87.3 | 92.7 | 95.7 | 97.7 | 98.8 |
| R50 + JCF ($D = 32, R = 8$) | 58.1 | 70.4 | 80.3 | 87.6 | 93.0 | 96.4 | 74.2 | 83.4 | 89.7 | 93.9 | 96.5 | 98.4 |

## 7 CONCLUSION

In this paper, we propose a new pooling scheme based which is both efficient in performances (rich representation) and in representation dimension (compact representation). This is thanks to the second-order information that allows richer representation than first-order statistics and thanks to a codebook strategy which pools only related features. To control the computational cost, we extend this pooling scheme with a factorization that shares sets of projections between each entry of the codebook, trading fewer parameters and fewer computation for a small loss in performance. We achieve state-of-the-art results on Stanford Online Products and Cars-196, two image retrieval datasets. Even if our tests are performed on image retrieval datasets, we believe our method can readily be used in place of global average pooling for any task.

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
