# OpenReview forum: "Efficient Codebook and Factorization for Second Order Representation Learning"
_ICLR.cc/2019/Conference_

### Official Review · AnonReviewer2 · 2018-11-01
**The proposed representation method is tested on only one task and considers only one evaluation metric**

**Rating:** 5
**Confidence:** 2

**Review:**

The paper proposes a second order method to represent images. More exactly, multiple (low-dimensional) projections of Kronecker products of low-dimensional representations are used to represent a limited set of dimensions of second-order representations. It is an extension of HPBP (Kim et al., ICLR 2017) but with codebook assigment.

The main advantage of the method is that, if the number of projection dimensions is small enough, the number of learned parameters is small and the learning process is fast. The method can be easily used as last layers of a neural network. Although the derivations of the method are straightforward, I think the paper is of interest for the computer vision community.

Nonetheless, I think that the experimental evaluation is weak. Indeed, the article only considers the specific problem of transfer learning and considers only one evaluation metric (recall@k). However, recent papers that evaluate their method for that task also use the Normalized Mutual Information (NMI) (e.g. [A,B]) or the F1-score [B] as evaluation metrics.
The paper does not compare the same task and datasets as (Kim et al., ICLR 2017) either.
It is then difficult to evaluate whether the proposed representation is useful only for the considered task. Other tasks and evaluation metrics should be considered.
Moreover, only the case where D=32 and R=8 are evaluated. It would be useful to observe the behavior of the approaches for different values of R.
In Section 3.2, it is mentioned that feature maps become rapidly intractable if the dimension of z is above 10. Other Factorizations are then proposed. How do these factorizations affect the second order nature of the representation of z? Is the proposed projection in Eq. (10) still a good approximation of the second order information induced by the features x?


The paper says that the method is efficient but does not mention training times. How does the method compare in terms of clockwork times compared to other approaches (on machines with similar architecture)?

In conclusion, the experimental evaluation of the method is currently too weak.


[A] Hyun Oh Song, Stefanie Jegelka, Vivek Rathod, Kevin Murphy: Deep Metric Learning via Facility Location. CVPR 2017
[B] Wang et al., Deep Metric Learning with Angular Loss, ICCV 2017

after rebuttal:
The authors still did not address my concern about testing on only one task with only one evaluation metric.

---

> ### Author Response · Authors · 2018-11-23
> **Additional experiments and method efficiency**
>
> First, we would like to thank you for the high-quality review.
>
> We answer to your concern about experimental evaluation and to questions on the efficiency of our method.
>
> Q) The experimental evaluation is weak.
> A) Since this was also pointed out by the other reviewers, we added section 5 about classification performances for fair and relevant comparison with state-of-the-art bilinear factorization methods on 3 fine-grained visual classification (FGVC) datasets (CUB, CARS and AIRCRAFT). For detailed information, we refer to the general comment and to the new version of the paper.
>
> We train our method and 4 similar approaches proposed by Reviewer3 to confirm the necessity of the codebook strategy to provide compact but informative representations. These methods are multi-rank extensions of the factorization in Eq.(6) with the improvement proposed in HPBP [Kim et al., 2017] and HPBP formulation that we have trained on classification tasks. The conclusion is that all multi-rank methods lead to the same performance as HPBP but with smaller representation while using the same number of parameters. However, JCF consistently outperforms these methods on the 3 datasets and leads to state-of-the-art results on 2 of them with much more compact representation (512 vs ~10k).
>
> These results confirm our intuition about the necessity to aggregate only similar features: The multi-rank methods do not have such selection mechanism preceding the projection, hence aggregate dissimilar features together. Moreover, the non-linearity not necessary improves performances as they are added after the projection is made. Although it is still possible to learn a projection coupled with the non-linearity to build a similarity driven aggregation, it would be not enforced by design. As JCF is designed for this similarity driven aggregation which is easier to train and lead to better results.
>
> Q) NMI and F1-score
> A) Due to time concern, we have chosen to focus on providing experimental results on newer datasets (CUB, CARS and AIRCRAFT) and for a different task (classification) for fair and relevant comparison with standard bilinear factorization methods, including HPBP [Kim et al., 2017].
>
> Q) The paper does not compare the same task and datasets as [Kim et al., 2017].
> A) We provide additional results on fine-grained visual classification datasets, including HPBP, with the same number of parameters as it is the standard task to evaluate bilinear representation. For more details, we refer you to the general comment and the new paper version.
>
> Q) Only the case D=32 and R=8 is evaluated
> A) In the original version of the paper, we provide ablation study of the number of clusters (Table 2) and the impact of the rank (Figure 1). Note that in the new version, Figure 1 has been replaced by Table 3. For the comparison to the state-of-the-art, we report the full model and the best compromise in performance loss/number of parameters from these ablation studies.
>
> Q) How do these factorizations affect the second order nature ?
> A) All enforced decompositions are only done on the projection matrix and not on the second order features. Thus, we can rebuild the original projection matrix which will be a sum of forth order tensors to project the original second order features.
>
> Q) How does the method compare in terms of clockwork time ?
> A) In terms of clockwork time, all methods implemented in Table 4 take roughly the same clockwork time as they can efficiently be implemented using standard layer in deep learning or optimized linear algebra computation. Quantitatively, the training (resp. testing) time for one epoch is around 300s (resp. 60s) using 448x448 images and batch of 32 (resp. 64), that is around ~ 50ms/image (resp. ~ 10ms/image). All of these computation times are given on CUB dataset, include the extraction of feature maps with VGG16 and are done on a single Nvidia GTX 1080 Ti.

---

> ### Author Response · Authors · 2018-12-11
> **Additional experiments on a different task**
>
> Dear reviewer,
>
> Remark that the revised paper includes new experiments on fine-grain classification (different from the original retrieval experiments) evaluated with the accuracy metric (different from the recall in the retrieval experiments) in the added Section 5 and Table 4, including references suggested by the AC. We believe these additional experiments fully address the concerns about testing on only one task with only one evaluation metric.

---

### Official Review · AnonReviewer3 · 2018-11-02
**Interesting idea of bilinear pooling with codebooks. But, needs more experiments for validating the idea.**

**Rating:** 6
**Confidence:** 4

**Review:**

Summary:
This paper proposes a novel bilinear representation based on a codebook model.
The proposed method is build upon the form of Hadamard Product for efficient representation of a bilinear model.
Through introducing the framework of codebook, it is naturally extended into a multiple-rank representation while the efficient pooling scheme is also derived from the (sparse) codeword assignment.
In addition, the authors also present an efficient formulation in which the codebook-based projections are factorized via a shared projection to further reduce the parameter size.
The experimental results on image retrieval tasks show that the proposed method produces better classification accuracy with a limited amount of parameters.

Comments:
Pros:
+ It is interesting that one-rank bilinear pooling is naturally extended to multiple-rank one via introducing codebooks.
+ Good performance in the image retrieval tasks.

Cons:
- This paper lacks important comparison for fairly evaluating the effectiveness of the proposed formulation.
- It also lacks detailed description and discussion for the methods.

Due to the above-mentioned weak points, the reviewer cannot fully understand whether the performance improvement really comes from the proposed formulation or not. Thus, this manuscript is currently judged as border. The detailed comments are shown in the followings.

- Comparison
Eventually, the proposed method is closely related to the multiple-rank representation of a bilinear model;

z_i = x^T W_i x (Eq.5) ~ x^T u_i v_i^T x (one-rank, Eq.6) ~ x^T U_i V_i^T x (multiple-rank), ... Eq.(A)

which is a straightforward extension from the one-rank model. From this viewpoint, the proposed form in Eq.10 is regarded as an extension of (A) by introducing non-linearity as

z_i = x^T U_i {h(x)h(x)^T} V_i^T x.  ... Eq.(10)

Thus, the main technical contribution is found in the weighting by {h(x)h(x)^T}, but its impact on the performance is not evaluated in the experiments. Furthermore, it is also possible to simply introduce such a non-linearity into the model (A) according to [Kim et al.,2017];

z_i = \sigma(x^T U_i) \sigma(V_i^T x) = 1^T {\sigma(U_i^T x) .* \sigma(V_i^T x)}, ... Eq.(B)

where ".*" indicates Hadamard Product, and we can more directly apply the method of [Kim et al., 2017] to the multiple-rank model by

z_i = p^T {\sigma(U_i^T x) .* \sigma(V_i^T x)}, ... Eq.(C)

where p is a R-dimensional vector. On the other hand, it is also necessary to compare the proposed method with [Kim et al.,2017] which is formulated by

z = P^T {\sigma(U^T x) .* \sigma(V^T x)}, ... Eq.(D)

where U and V are matrices of d x K and P is K x D. The parameter K (shared rank) should be determined so that the total parameter size of (2dK + KD) is compatible to that of the proposed method, 2NdD.

In summary, for demonstrating the effectiveness of the proposed method in Eq.(10), it is inevitable to compare it with the models (A, B, D) and hopefully (C).

- Presentation
In Section 4.2, the performance results of the factorization model in Eq.(13) are merely shown without deep discussion nor analysis on them. In particular, it is unclear why the JCF of N=32 and R=32 outperforms the CHPBP of N=32. Those two methods are different only in the form of U and V:
(CHPBP) U_i -> U'_i A (JCF),
where U_i and U'_i have the same dimensionality of d x 32, and thus we can say that JCF overly parameterizes the projection by redundantly introducing A of 32 x 32. Thus, the projection capacity of JCF is completely the same as that of CHPBP. Therefore, it needs detailed discussion for the performance improvement shown in Figure 1.

Minor comments are:
* There are lots of notations, and thus it would be better to show a summary of the notations.
* In Section 4.1, there is no clear description about the methods of BP and HPBP. Actually, the method of HPBP is different from the one presented in [Kim et al., 2017].

---

> ### Author Response · Authors · 2018-11-23
> **Additional comparisons and discussion**
>
> We would like to thank you for your high-quality review and the insightful propositions to strengthen our paper.
>
> We answer to both your concerns on lack of comparison and detailed description/discussion.
>
> 1) Lack of comparison
> Also requested by the other reviewers, we report in table 4 classification accuracy on 3 fine-grained visual classification (FGVC) datasets: CUB and CARS with their respective split for classification and AIRCRAFT. We compare our method to the state-of-the-art on Bilinear factorization schemes to confirm the advantage of the codebook strategy to produce compact but rich representations. We report comparable results to the state-of-the-art with much more compact representations than others.
>
> Moreover, we also follow your proposed comparison and we implement and train the 4 approaches, that is : HPBP from [Kim et al., 2017], the multi-rank extension of the standard compression and the improvement proposed in HPBP to improve the multi-rank formulation (non-linearity and projection). Results and discussion are available in Table 4 from the new paper version.
>
> 2) Lack of discussion
> We add discussion of these results at the end of the section 5. To sum-up, HPBP and multi-rank methods in respectively 2048 dimensions and 512 dimensions are slightly lower than the state-of-the-art (~1%) while our JCF consistently outperforms these multi-rank formulations. This confirm our intuition about the importance of aggregating and projecting only related features together. Moreover, the non-linear multi-rank variants are harder to train and may not bring improvements. These multi-rank strategies do not have a selection mechanism that projects and then aggregates only related features. Moreover, the non-linearity leads to models that may be efficient, but harder to train in practice. Although it is still possible to learn a projection coupled with the non-linearity that would lead to a similarity driven aggregation, it is not enforced by design. The advantage of JCF is to provide by design this similarity driven aggregation, which leads to better results.
>
> 3) Performances of JCF(32, 32) vs C-CBP(32)
> We argue that such variation in performances are usually induced by the difficulty in practice to learn the codebook. For example, C-CBP with a 32 size codebook may in practice only use in average 28 entries, which leads to a sub-efficient representation. This is mainly due to the implicit learning of the codebook (contrarily to the cross-entropy loss used directly in classification for example). Thus, the over-parametrization allows the use of these additional entries to improve its representations.

---

### Official Review · AnonReviewer1 · 2018-11-02
**Method needs some clarification.**

**Rating:** 4
**Confidence:** 5

**Review:**

Summary: This paper presents a way to combine existing factorized second order representations with a codebook style hard assignment. The number of parameters required to produce this encoded representation is shown to be very low. Like other factorized representations, the number of computations as well as the size of any intermediate representations is low. The overall embedding is trained for retrieval using a triplet loss. Results are shown on Stanford online, CUB and Cars-196 datasets.

Comments:

Review of relevant works seems adequate. The results seem reproducible.

The only contribution of this paper is combining the factorized second order representations  of (Kim et. al. 2017) with a codebook style assignment (sec. 3.2). Seems marginal.

The scheme described in Sec. 3.2 needs clarification. The assignment is applied to x as h(x) \kron x in (7). Then the entire N^2 D^2 dimensional second order descriptor h(x) \kron x \kron h(x) \kron x is projected on a N^2 D^2 dim w_i. The latter is factorized into p_i, q_i \in \mathbb{R}^{Nd}, which are further factorized into codebook specific projections u_{i,j}, v_{i,j} \in \mathbb{R}^{d}. Is this different from classical assignment, where x is hard assigned to one of the N codewords as h(x), then projected using \mathbb{R}^d dimensional p_i, q_i specific to that codeword ?

In section 4.1 and Table 2, is the HPBP with codebook the same as the proposed CHPBP ? The wording in "Then we re-implement ... naively to a codebook strategy"  seems confusing.

The method denoted "Margin" in Table 4 seems to be better than the proposed approach on CUB. How does it compare in terms of efficiency, memory/computation ?

Is it possible to see any classification results? Most of the relevant second order embeddings have been evaluated in that setting.


===============After rebuttal ===============================

After reading all reviews, considering author rebuttal and AC inputs, I believe my initial rating is a bit generous. I would like to downgrade it to 4. It has been pointed out that many recent works that are of a similar flavor, published in CVPR 2018 and ECCV 2018, have slightly better results on the same dataset. Further, the only novelty of this work is the proposed factorization and not the encoding scheme. This alone is not sufficient to merit acceptance.

---

> ### Author Response · Authors · 2018-11-23
> **Contributions, clarifications and classification experiments**
>
> We would like to thank you for the high-quality review.
>
> We clarify our contributions and the pointed formulation. Finally, we also report classification results.
>
> Q) The only contribution of this paper… Seems marginal.
> A) We are convinced by the novelty of this paper, as the few methods that combined second order representations and codebook strategies faced the very high dimensionality of the representation without proposing compact factorization schemes (as shown in Table 1). E.g. MFAFVNet (Li et al., 2017b) extend this second order pooling to codebook with factorization scheme. However, even with such factorization, their proposed method leads to 500k dimension representations (i.e. 2x  Bilinear Pooling) , 27M parameters and  around 40GOps to compute the output representation. JCF provides 512d representation (1/1000 factor), 4M parameters (1/7) and 3GOps (1/10) with the same performances as well-known second order factorization scheme.
>
> To sum up: Combining second order representation with a codebook is not trivial due to computation concerns. We are the first to provide an efficient scheme for such combination.
>
> Q) Is this different from classical assignment ?
> A) Without the introduction of rank factorization, there is no difference: the two projections u_{i,j} and v_{i,j} play a similar roles as intra-projection in VLAD representation for example. This insight is developed in section 3.2 between equations (9) and (10). However this approach is not tractable, which is the reason we proposed our joint factorization and codebook method. Moreover, to make the method end-to-end trainable, x is soft-assigned to the codebook instead of the hard-assignment of VLAD.
>
>
> Q) The wording in “Then we re-implement...naively to a codebook strategy” seems confusing.
> A) Thanks to point out this ambiguity, we update this sentence in the new paper version also following “Reviewer3” remark on the terms used.
> For clarity, we implement Bilinear Pooling (BP) and  Compact Bilinear Pooling (CBP) and not HPBP as we do not add the non-linearity nor the projection. However, we also train the projection matrices for CBP. Then, we extend these two methods naively to codebook strategy, that is by computing:
> - W^T[h(x) \kron x \kron x] where W \in \mathbb{R}^{Nd^2 \times D} (named C-BP)
> -  CBP extended to codebook (named C-CBP), using equation (10).
>
> Q) “Margin” in Table 4 performs better on CUB, how does it compare ?
> A) The “Margin” method proposed a new sampling strategy that allows to explore much more informative triplet than standard strategies, including ours. Also, they use larger images. Note that we can exploit their sampling method to improve our training procedure.
>
> Q) Is it possible to see any classification results ?
> A) As mentioned in the general comment, we added results on 3 fine-grained visual classification (FGVC) datasets, CUB and CARS using the standard split in FGVC and Aircraft which are 3 common datasets for FGVC task. We compare our method to other factorization scheme and we report comparable results to the state-of-the-art with much more compact representation. For more details, we invite you to read the general comment and the new paper version.

---

### Author Response · Authors · 2018-11-23
**Additional classification experiments**

First, we would like to thank our three reviewers who provided high-quality reviews and insightful remarks, but also proposed additional experiments to strengthen this paper.

In this comment, we answer to the major concern of this paper: fair and relevant comparison.

We agree with the 3 reviewers about the lack of comparison on fine-grained visual classification datasets (FGVC) against other factorization scheme. Hence, we add section 5 to discuss about the performances of our method on FGVC task.

We provide an overview of this additional section in this comment.

To evaluate the compactness of our method but also its efficiency, we compare our method on three FGVC datasets named CUB, CARS and AIRCRAFT. We report the results of the 4 equations proposed by “Reviewer 3”, that is : (I) HPBP from [Kim et al., 2017] ; (II) the extension of Eq.(6) to multi-rank ; (III) the introduction of the same non-linearity as HPBP in the multi-rank formulation and (IV) with an additional learned combination of ranks.

All multi-rank approaches are computed with a rank of 8 and a representation dimension of 512 while HPBP uses 2048 dimensions (to be fair in number of parameters). We compare these methods to JCF(N=32, R=8).

Our method consistently outperforms the multi-rank variants. This confirms our intuition about the importance of grouping features by similarity before projection and aggregation. Indeed, multi-rank variants do not have a selection mechanism preceding the projection into the subspace that would allow to selectively choose the projectors based on the input features. Instead, all features are projected using the same projectors and then aggregated. We argue that non-linear multi-rank variants bring only marginal improvements, since the non-linearity happens after the projection is made. Although it is still possible to learn a projection coupled with the non-linearity that would lead to a similarity driven aggregation, it is not enforced by design. Since JCF does the similarity driven aggregation by design, it is easier to train, which we believe explains the results.

The code for our model and the additional multi-rank experiments will be released after the reviewing process.

---

### Comment · Area_Chair1 · 2018-11-26
**Comparison with other recent works.**

Dear authors and reviewers.

At a glance, some of the missing recent works which outperform the proposed scheme are missing. For instance compact representations:
[1] MoNet: Moments Embedding Network by Gou et al. (e.g. Stanford Cars via Tensor Sketching: 90.8 vs. 90.4 in this submission, Airplane: 88.1 vs. 87.3% in this submission, 85.7 vs. 84.3% in this submission)
[2] Second-order Democratic Aggregation by Lin et al. (e.g. Stanford Cars: 90.8 vs. 90.4 in this submission)
[3] Statistically-motivated Second-order Pooling by Yu et al (CUB: 85%)

Also, regular Bilinear representations:
[4] DeepKSPD: Learning Kernel-matrix-based SPD Representation for Fine-grained Image Recognition by Engin et al.

Moreover, it seems that basic Bilinear pooling (although non-compact) reaches 86.4, 89.3 and 91.8% on CUB, Airplane, and Cars again outperforming the results achieved in this paper.

Approaches such as [4] go even further with Bilinear pooling: 93.2% for Cars vs. 90.4% in this submission, 91.5% for Aircraft.

Given Bilinear pooling is a very crowded topic in venues such as CVPR, ECCV and ICCV, it begs a question whether the proposed here compact representations are really a non-incremental non-trivial achievement that builds on the Bilinear pooling. It feels as the authors avoid comparisons with some recent very strong baselines. From my own knowledge, recent CVPR'18 and ECCV'18 conferences published at least 6 new papers on this topic; non of which seems to be referenced in this manuscript. Moreover, [3] can challenge the proposed here approach in terms of compactness, [2] in terms of compactness and speed, [4] mainly in terms of excellent results via Bilinear strategy.

From minor comments, equations such as Eq. 3 (mere expansion of a polynomial kernel) appeared in many works preceding Gao et al. (2016) backdating to year 2000 or so. Perhaps citing Gao et al. next to such an expansion is not really the most factual take on the prior literature. Dedicating Eq. 1, 2 and 3 to a simple polynomial expansion that ends up being an outer product also feels as perhaps unnecessary and can be left out for a supplementary material.

---

> ### Author Response · Authors · 2018-11-26
> **Comparison with other recent works**
>
>
> We would like to thank you for pointing out these recent references. However, most of them do not focus on compactness and efficiency which is the main focus of this paper and consequently do not offer a fair comparison. We aim at low dimensional features and low complexity, while they focus on improved accuracy regardless of the cost. Recall that we also report very good results on retrieval tasks and datasets, for which compactness is an essential property.
>
> As a sum-up, references [2, 4] use 262k dimensions compared to our 512dimensions. [1] uses the same factorization as CBP-TS upon which we show improvements. Only [3] offers a fair comparison and achieves similar results as ours.
>
> In details:
>
> [1] MoNet: MoNet proposes a normalization using Tensor Sketch factorization with 10k dimensions to reach the reported performances. Furthermore, MoNet requires the computation of the SVD during the forward pass, which is a costly operation not comparable with a feed forward only implementation.
>
> [2] Second-order democratic aggregation: The reported results with VGG16 use 262k dimensions. Moreover, the forward pass relies on the Dempened Sinkhorn Algorithm, which is an iterative algorithm and thus not comparable with simple feed forward methods.
>
> [3] Statistically motivated second order pooling: The results reported with VGG16 on CUB are 85% at 2k dimensions and 82.6% at 64 dimensions. We report 84.3% at 512 dimensions, which we believe is comparable to their results assuming the usual logarithmic progression of accuracy versus dimension.
>
> [4] DeepKSPD: The reported results are obtained using the full bilinear dimension (262k) and are thus not comparable to compact representations. Moreover, they use a costly decomposition of the kernel matrix which means that neither compactness (as the full matrix has to be computed) nor efficiency can be achieved using this method.
>
>
> About "whether the proposed here compact representations are really a non-incremental non-trivial achievement that builds on the Bilinear pooling":
>
> We are to our knowledge the first to introduce an efficient and compact codebook strategy to bilinear models thanks to a non-trivial factorization scheme, with competitive results. All recently published methods focus on improving the accuracy (for example by proposing better normalization), but are consequently not compatible with
> compact factorization that make bilinear models tractable. As such, we believe our method offers a nice accuracy improvement to compact and efficient bilinear models, which recently published methods are not able to do due to their lack of factorization.
>
>
> About the minor comment:
>
> The goal of equations 1 and 2 is mainly to introduce our notations and the tensor framework to ease the transition to section 3. Eq.(3) has the same objective: Presenting this known expansion to ease the transition from Eq.(5) to Eq.(6) and from Eq.(7) to Eq.(8). We do nonetheless agree that these are common equations, and we add them only for the sake of clarity.
>
> Edit: We upload a new version of the paper with references [1] and [3] added to Table 4.

---

### Meta-Review · Area_Chair1 · 2018-12-15
**Incomplete work.**

**Confidence:** 5
**Recommendation:** Reject

**Metareview:**

AR1 is is concerned that the only contribution of this work is  combining second-order pooling with with a codebook style assignments. After discussions, AR1 still maintains that that the proposed factorization is a marginal contribution. AR2 feels that the proposed paper is highly related to numerous current works (e.g. mostly a mixture of existing contributions) and that evaluations have not been improved. AR3 also points that this paper lacks important comparisons for fairly evaluating the effectiveness of the proposed formulation and it lacks detailed description and discussion for the methods.

AC has also pointed several works to the authors which are highly related (but by no means this is not an exhaustive list and authors need to explore google scholar to retrieve more relevant papers than the listed ones):

[1] MoNet: Moments Embedding Network by Gou et al. (e.g. Stanford Cars via Tensor Sketching: 90.8 vs. 90.4 in this submission, Airplane: 88.1 vs. 87.3% in this submission, 85.7 vs. 84.3% in this submission)
[2] Second-order Democratic Aggregation by Lin et al. (e.g. Stanford Cars: 90.8 vs. 90.4 in this submission)
[3] Statistically-motivated Second-order Pooling by Yu et al (CUB: 85%)
[4] DeepKSPD: Learning Kernel-matrix-based SPD Representation for Fine-grained Image Recognition by Engin et al.
[5] Global Gated Mixture of Second-order Pooling for Improving Deep Convolutional Neural Networks by Q. Wang et al. (512D representations)
[6] Low-rank Bilinear Pooling for Fine-Grained Classification' by S. Kong et al. (CVPR I believe). They get some reduction of size of 10x less than tensor sketch, higher results than here by some >2% (CUB), and all this obtained in somewhat more sophisticated way.

The authors brushed under the carpet some comparisons. Some methods above are simply better performing even if cited, e.g. MoNet [1] uses sketching and seems a better performer on several datasets, see [2] that uses sketching (Section 4.4), see [5] which also generates compact representation (8K). [4] may be not compact but the whole point is to compare compact methods with non-compact second-order ones too (e.g. small performance loss for compact methods is OK but big loss warrants a question whether they are still useful). Approach [6] seems to also obtain better results on some sets (common testbed comparisons are essentially encouraged).

At this point, AC will also point authors to sparse coding methods on matrices (bilinear) and tensors (higher-order) from years 2013-2018 (TPAMI, CVPR, ECCV, ICCV, etc.). These all methods can produce compact representations (512 to 10K or so) of bilinear or higher-order descriptors for classification. This manuscript fails to mention this family of methods.

For a paper to be improved for the future, the authors should consider the following:
- make a thorough comparison with existing second-order/bilinear methods in the common testbed (most of the codes are out there on-line)
- the authors should vary the size of representation (from 512 to 8K or more) and plot this against accuracy
- the authors should provide theoretical discussion and guarantees on the quality of their low-rank approximations (e.g. sketching has clear bounds on its approximation quality, rates, computational cost). The authors should provide some bounds on the loss of information in the proposed method.
- authors should discuss the theoretical complexity of proposed method (and other methods in the literature)

Additionally, the authors should improve their references and the story line. Citing  (Lin et al. (2015)) in Eq. 1 and 2 as if they are the father of bilinear pooling is misleading. Citing (Gao et al. (2016)) in the context of polynomial kernel approximation in Eq. 3 to obtain bilinear pooling should be preceded with earlier works that expand polynomial kernel to obtain bilieanr pooling. AC can think of at least two papers from 2012/2013 which do derive bilinear pooling and could be cited here instead. AC encourages the authors to revise their references and story behind bilinear pooling to give unsuspected readers a full/honest story of bilinear representations and compact methods (whether they are branded as compact or just use sketching etc., whether they use dictionaries or low-rank representations).

In conclusion, it feels this manuscript is not ready for publication with ICLR and requires a major revision. However, there is some merit in the proposed direction and authors are encouraged to explore further.